# Application of Multi-Criteria Decision-Making Analysis to Rural Spatial Sustainability Evaluation: A Systematic Review

**DOI:** 10.3390/ijerph19116572

**Published:** 2022-05-27

**Authors:** Zheng Yuan, Baohua Wen, Cheng He, Jin Zhou, Zhonghua Zhou, Feng Xu

**Affiliations:** 1School of Architecture and Planning, Hunan University, Changsha 410082, China; yuanzheng0416@hnu.edu.cn (Z.Y.); wenbaohua@hnu.edu.cn (B.W.); hecheng2018@hnu.edu.cn (C.H.); zhoujin_y@hnu.edu.cn (J.Z.); zhouzh@hnu.edu.cn (Z.Z.); 2Hunan Key Laboratory of Sciences of Urban and Rural Human Settlements at Hilly Areas, Changsha 410082, China

**Keywords:** multi-criteria decision-making, rural areas, spatial evaluation, sustainability, research review

## Abstract

The rational allocation of spatial resources is an important factor to ensure the sustainable development of rural areas, and effective pre-emptive spatial evaluation is the prerequisite for identifying the predicament of rural resource allocation. Multi-criteria decision-making analysis has advantages in solving multi-attribute and multi-objective decision-making problems, and has been used in sustainability evaluation research in various disciplines in recent years. Previous studies have proved the value of spatial evaluation using multi-criteria decision analysis in guiding rural incremental development and inventory updates, but systematic reviews of the previous literature from a multidisciplinary perspective and studies of the implementation steps of the evaluation framework are lacking. In the current paper, the research is reviewed from the two levels of quantitative statistics and research content, and through vertical and horizontal comparisons based on three common operating procedures: standard formulation, weight distribution, and ranking and verification. Through the results, the application status and characteristics of the MCDA method in related research are determined, and five research foci in the future are proposed.

## 1. Introduction

### 1.1. Background Overview

Driven by the global environmental crisis, sustainability has gradually become the focus of rural development. The ultimate goal of sustainability can be seen as a balance between society, the economy, and the environment [1]. Rural sustainable development can be considered as a process of “sustainable governance of resources” [2]. Based on the urgent needs of sustainability, the rational allocation of production, living, and ecological resources is of great significance to promoting sustainable rural development [3]. As the provider of these resources, space is the most important resource to support sustainable rural development [4]. Therefore, scientific spatial evaluation is a prerequisite for ensuring the rational allocation of other rural resources.

The evaluation of rural spatial sustainability refers to the classification of research problems. That is, with the goal of sustainable development and the spatial resources of rural areas as the evaluation object, it aims to improve one or more aspects of rural sustainability by identifying and even improving the spatial layout. Generally, existing rural spatial evaluations are closely related to the term “spatial decision-making”, which in turn guides the selection of subsequent programs or strategies [5]. The existing evaluation of rural space, aiming at sustainability, involves many aspects, such as natural resources [6,7], public facilities [8], energy services [9], and planning and design [10,11]. Previous studies have demonstrated that rural-related spatial evaluation or decision making is a complex task [12]. Therefore, achieving sustainability goals also requires new approaches to decision making based on knowledge production [13]. Over the past two decades, multi-criteria decision-making analysis methods have been applied in related fields to assist decision makers. Multi-criteria decision-making analysis may be an effective and efficient option for solving such problems.

### 1.2. Multi-Criteria Decision-Making

Multi-criteria decision making (or multi-criteria decision-making analysis, hereafter referred to as MCDA) is a basic scientific method used to select the best alternative, classify alternatives, or rank alternatives in order of preference [14,15]. The term MCDM is used for all methods and techniques that involve multiple conflicting criteria used by decision makers to derive preferences [14]. MCDA originated in the field of operations research. After evolution and development, MCDA has been widely used in management science [16,17], environmental science [18,19], geography [18,20], material science [21], energy [22], mathematics [23], information and computer science [24], etc. In recent years, MCDA methods have been used to assist decision making related to sustainability [25]. To date, more than 100 MCDA methods have been used in practice, including original methods, derived methods, and combined methods [26], but only a few MCDA methods have been applied to solving rural spatial evaluation problems.

### 1.3. Literature Review

Table 1 summarizes the details of two reviews with similarities and correlations to this study.

Gebre et al. [27] provided a quantitative and descriptive review of the application of the MCDA approach to solving sustainable land allocation problems in rural areas. The study screened 69 articles from Web of Science, Science Direct, and Google Scholar for review. Quantitative and qualitative overviews were carried out on the basis of the categorized research problems. This review only focused on the research field of land use. It was not a comprehensive review of all disciplines. Secondly, some of the literature in this review dealt with policy decisions, and not all of them focused on the spatial issue of land allocation.

Mudashiru et al. [28] provided a quantitative and narrative review of the application of MCDA, statistics, and machine learning methods in sustainable planning and management to identify flood hazard-prone areas in rural areas. The review screened 84 articles from Science Direct and Scopus, of which 46 used the MCDA method. This review provided an overview of the application of different kinds of quantitative or semi-quantitative methods using a cluster analysis approach. However, the review only focused on the planning and management of flood hazards. Secondly, the review did not take into account the huge amount of data in the Web of Science database, and the duplication of Science Direct and Scopus may not be conducive to expanding the relevant literature. Finally, the study only summarized the application of MCDA with a descriptive narrative.

After reviewing the limitations and distinctions of the above reviews, we believe that a comprehensive review of the literature related to the topic of this study is warranted to provide detailed insights.

### 1.4. Research Purpose

In view of the fact that the application of the MCDA method to solve the problem of rural spatial sustainability evaluation is not clear, this review aims to:(1)Provide an up-to-date overview of the application of the MCDA method to the issue of rural spatial sustainability evaluation;(2)Determine the specific research topic or unit ranges for which these MCDA methods are commonly used;(3)Identify the criteria aspects most commonly considered in the MCDA process;(4)Determine the feature and distribution of MCDA methods in each operation stage in terms of time and research topics;(5)Clarify the focus of future research work.

This review will be useful in terms of the following aspects. Firstly, this review will help researchers and practitioners gain a detailed understanding of this research topic. Secondly, it introduces the commonly used MCDA methods in the research field, and provides guidance on the selection tendency of these methods.

The paper consists of five sections as follows. In the “Methods and Data Sources” section, the classification of MCDA methods, the structure of the study, the sources of data, and the screening methods are introduced. In the “Primary Review Analysis and Results” section, the overview and trends of the literature are introduced, and several research topics and scopes are clarified. In “Detail Review Analysis and Results”, the application of MCDA methods in three stages of operation is analyzed. In “Discussion”, the results of the above two sections are further refined and fully discussed. The last section is the “Conclusion” of the research.

## 2. Methods and Data Sources

### 2.1. Classification of MCDA Methods

As a comprehensive evaluation method, MCDA is suitable for solving complex problems characterized by uncertainty, conflicting goals, information heterogeneity, and diversity of interests or viewpoints, in addition to explaining dynamic development systems under the influence of multiple factors [27]. In contrast to the traditional single-criteria analysis method, the MCDA method has advantages because it can obtain the comprehensive score or the ranking of the objects under the influence of multiple criteria [29]. From the perspective of the attribute of the judgment subject, MCDA methods can be divided into three categories: multi-objective decision-making analysis (MODA), multi-attribute decision-making analysis (MADA), and comprehensive MCDA [30]. From the perspective of the operation process and system composition, Figure 1 clearly shows the six operational steps and the three sub-systems of the MCDA method. At present, there is no unified classification standard for the criteria, but evaluation research aiming at sustainability usually divides the criteria into three categories: society, the economy, and the environment [31,32]. The index is used as a quantitative basis for measuring the quality of the criteria. From the perspective of using functions, Table 2 divides the MCDA methods commonly used in sustainability evaluation into weighting methods and ranking methods, and some methods have both functions [14,25,30,31,33,34]. The weighting methods can be divided into the subjective weighting method and objective weighting method. The subjective weighting method depends only on the preference of the decision maker, and not on the quantitative measured data of the evaluation target. In contrast, the objective weighting method is obtained mathematically through the analysis of the initial data [31,35]. Both methods have their advantages and disadvantages. Subjective weighting is more explanatory and misleading, and objective weighting is resistant to interference, but sometimes exceeds the limits of research assumptions. Usually, in a complete multi-criteria decision-making process, at least one weighting method and ranking method are required.

### 2.2. Research Structure

Figure 2 presents the five main parts of this study and the sub-steps in each part. After the literature data were obtained and the research sample was determined, primary and detailed review analysis were carried out. Bibliometric methods have been widely and commonly used to evaluate and review existing research, and the analysis scope includes publication pattern research, bibliographic compilation, bibliographic coupling (co-citation and co-occurrence), and citation analysis (scientific papers and patents) [64]. This study used VOSviewer 1.6.17 as an auxiliary tool for data processing in the primary review. Finally, according to the three operating procedures of criteria and index, weight and priority, and ranking and verification, the detailed review of the sample was analyzed from the vertical and horizontal perspectives, including a key literature commentary.

### 2.3. Bibliographic Search Engine Databases

This study selected five potential databases and identified their basic information [27,65,66,67]. Web of Science core collections (WOS), Scopus, Science Direct (SD), and Google Scholar (GS) are currently widely used literature databases. In addition, China’s rural area in 2021 is 8,868,577 square meters, and the rural population is 544 million, accounting for 7.93% and 16% of the world, and ranking second and third in the world, respectively. Its cereal production land comprises more than 100 million hectares, ranking first in the world [68]. Therefore, as the most important agricultural country in the world, the research progress of China’s rural spatial sustainability evaluation deserves attention. As the largest academic database in China, China National Knowledge Infrastructure (CNKI) was considered for this review. Table 3 clearly illustrates the characteristics of each database for better understanding.

Through the above comparison, WOS, Scopus, and CNKI were selected as the final data sources for this study because of their relatively large database capacity and powerful search function. Science Direct was abandoned due to poor search functionality and high overlap with Scopus data. Google Scholar was excluded due to the singularity of the search function, the lack of categorical tags, and imprecise search results [69,70].

### 2.4. Data Extraction and Procedures

This study used the PRISMA framework to identify the target literature for analysis. Previous studies have demonstrated the usefulness of the PRISMA framework for literature or data screening and further meta-analysis [71,72]. In terms of operation steps, it is mainly divided into four steps: Identification, Screening, Eligibility, and Included [71].

Step 1—Identification: This study used “rural” + “spatial” + “evaluation” + “sustainable”, and the combination between them: “rural sustainability” + “spatial evaluation” or “rural” + “spatial sustainability” + “evaluation”, as topics and keywords in CNKI, Scopus, and Web of Science (WOS) Core Collection to search academic journals and conference papers before 31 December 2021.

Step 2—Screening: After removing duplicates, the above databases yielded 741, 186, and 113 articles, respectively. Then, the results were refined by “multi-criteria decision-making”, “multi-criteria decision-making analysis”, and the names of MCDA methods in Table 2 to filter the evaluation literature using MCDA methods. This search resulted in 186 entries. 

Step 3—Eligibility: Further screening was based on the following principles to eliminate irrelevant entries:(1)The definition of the keyword “rural” includes documents that take the suburban, urban–rural border, township, and rural areas as the research scope, and the documents that take the city as the research object alone were not included.(2)The “space” referred to in this article is the concept of physical space in the category of geography, so the “space” of abstract content involving the universe, network, cogitation, and mathematics is not included in the scope of consideration. Furthermore, the space must be the evaluation object of the research, and the relevant literature must take the exploration of the spatial characteristics as the main purpose. If a study only referred to the research area of the geospatial attribute, and did not make space the evaluation object, it was not included.(3)Sustainability should be the research target of the considered literature. These studies should identify or improve one or more aspects of rural sustainability through spatial evaluation using the MCDA approach.

Step 4—Included: After the above screening, 103 documents were identified as the final sample for qualitative and quantitative analysis in this study.

## 3. Primary Review Analysis and Results

### 3.1. Journal and Publication Year

The amount of research on rural spatial sustainability evaluation using the MCDA method has increased over time. Figure 3a shows that the number of relevant publications surged in 2015 and 2018, and showed a steady upward trend in other years. In terms of publication years, the overall trend has increased since 2002. Figure 3b shows the publications and citations of all journals. The number of related papers published in *Land Use Policy* is the largest. Through statistical analysis of the published journals of the sample literature, it was found that the *Land Use Policy* journal published seven related publications. By calculating the total citations of the journals to which the sample papers belonged, it can be found that *Land Use Policy* still has the highest number, i.e., 334, so it can be considered to be an important journal for this theme. 

### 3.2. Author, Organization and Country/Area

The geographical distribution of the study area is characterized by a greater number of studies in the Eastern Hemisphere, fewer in the Western Hemisphere, more in the Northern Hemisphere, and fewer in the Southern Hemisphere. The research institutions and personnel are mainly concentrated in Asia and Europe. Since the CNKI database mainly includes research documents of authors and institutions in China, in order to objectively show the global distribution of the research areas in the relevant literature related to the research topic, the global distribution statistics of the 67 English documents from WOS and Scopus were determined. The global distribution of the relevant literature is shown below (Figure 4). It can be seen that China, Turkey, and Italy are the top three countries in terms of quantity; Iran and Spain are close behind. According to the number of publications and citations, Table 4 shows that six important authors and four research institutions in related fields were identified.

### 3.3. Keywords and Research Area

The 103 articles covered 26 research areas, four categories of keywords, and six research topics. Figure 5 shows all the research areas covered by the sample and the number of publications in each field. Environmental Sciences and Ecology is the category having the most publications. Through the co-occurrence analysis of the keywords, it can be seen that the most common keywords in the rural spatial evaluation research using the MCDA method are “Environmental science”, “GIS”, and “Science & Technology”. In order to compare the sample horizontally, it is a common to categorize topics according to the research field, content, problem, and goal [27,73]. Using VOSviewer 1.6.17 software to perform cluster analysis on keywords having a frequency of at least four, four categories were obtained (Figure 6). According to the content of keywords in the clustering, the above four categories can be defined as domain, target, problem, and method. In order to further determine the research focus, the above four categories of keywords and their related literature were screened, and six research topics were finally determined. Table 5 shows the names of the six research topics and the number of studies in each topic.

### 3.4. Unit Range

From the perspective of the unit range delineated by the rural space evaluation, all articles could be divided into four scale categories: Regional scale (36.89%), District/county area (44.66%), Community/village (13.59%), and Architecture (2.91%). Figure 7 shows the correlation characteristics between publication year, unit range, and research topic. The number of the related studies in Regional scale was relatively even in the past 10 years, whereas the studies involving District/county area and Community/village increased significantly in the past five years. With the exception of Habitat, which had the lowest number, and the Land use subject, which had the highest number, the Regional scale research had a relatively even distribution among the other four types of topic. With the exception of Conservation and Habitat, the distribution in District/county area is relatively even in the other four categories. Community/village research is involved in all six types of research topic, of which Habitat is the most distributed. Architecture research is only related to Conservation and Habitat.

### 3.5. MCDA Methods

Compared with other MCDA methods, AHP is the most commonly used. In this study, the clustering statistics along the time axis were carried out by the MCDA methods used in the three stages of the sample documents. Figure 8 shows that SAW was the earliest used MCDA method. Subsequently, the AHP method was introduced for the first time in the sustainable evaluation of rural space in 2005, and the number of related publications has increased over time, especially in the past five years. PCM is ranked second, and Delphi and TOPSIS follow. Several other MCDA methods have been gradually applied in rural spatial sustainability evaluation research in the past 10 years. In particular, in the last four years, qualitative analysis methods such as SWOT and TOWS have also begun to emerge, in addition to the traditional MCDA methods that rely on statistics for quantitative analysis.

## 4. Detail Review Analysis and Results

### 4.1. Criteria and Index

According to the ultimate goal of sustainability [1], the criteria system of most rural spatial sustainability evaluation studies includes the three categories of social, economic, and environmental content. In order to analyze the criteria components of the sample literature, the following equations were used to calculate the number of criteria in the evaluation system for the relevant literature in each year and each research topic.

Step 1: Calculate the respective proportions of social, economic, environmental, or other criteria in a single article:(1)$$P_{S,E\mathrm{co},E,O}=\frac{C_{S,E\mathrm{co},E,O}}{C_{T}}$$

In Equation (1), $C_{S,E\mathrm{co},E,O}$ is the number of criteria for society, economy, environment, or other in a single article, and $C_{T}$ is the total number of criteria in a single article.

Step 2: Calculate the arithmetic mean of the proportion of social, economic, environmental, or other criteria in the total *n* literature in the *j*th year: (2)$$A_{S,Eco,E,O}=\frac{\sum_{i=1}^{n}P_{S,Eco,E,O}}{n}$$

Step 3: Calculate the proportion of social, economic, environmental, or other criteria in the total m articles in a single research topic:(3)$$A_{S,E\mathrm{co},E,O}{}^{\prime }=\frac{\sum_{i=1}^{m}C_{S,E\mathrm{co},E,O}}{C_{T}^{\prime }}$$

In Equation (3), $C_{T}^{\prime }$ is the total criteria number of all studies in the corresponding research topic.

#### 4.1.1. The Formulation of Criteria

##### Quantitative Analysis of Criteria

Although the earliest published literature in the sample is from 2002, since the literature before 2010 was mainly based on theoretical research and did not include empirical research on specific criteria data, 2010 was considered the starting year. 

Figure 9 indicates that the number of criteria for the three categories of society, economy, and environment has generally experienced a slightly fluctuating upward trend since 2010. The overall number of environmental criteria is larger than that of social and economic criteria, and the number of social criteria is slightly higher than that of economic criteria. However, with the enrichment in research topics, the difference in research focus has led to subsequent fluctuations in the distribution of social, economic, and environmental criteria. 

Figure 10 shows that the number of criteria at the environmental category still accounts for a large proportion. With the exception of Tourism, the remaining five categories of topics all exceed 30%. The overall proportion of criteria at the economic category is still relatively low, especially in terms of Site selection, Tourism, Conservation, and Habitat. The social criteria accounted for the highest proportion in Habitat; Site selection and Tourism ranked second.

##### Content Analysis of Criteria

Firstly, the measurement angle of environmental criteria is the most multi-dimensional factor. The element composition of the environment can be divided into atmosphere, water, soil, geological, and biological environments. The inherent characteristics of the environment can be classified into three types: capacity, stability, and complexity. Therefore, necessity and diversity lead to dominance of environmental criteria. Secondly, a number of studies do not strictly distinguish the three categories of society, economy, and environment to construct a criteria system. One of the most common approaches is to combine social and economic criteria [78,89,95,116,144,149,160]. Finally, the evaluation steps of criteria are not single. Some studies regarding Site selection set up a multi-stage index system [20,110,158], by setting one or more constraint criteria systems as the access rules for screening samples before the spatial sustainability evaluation. The content involved in other criteria of each topic are shown in Table 6. In addition, Rahmoun et al. [138], starting from the concept of “peaceful tourism”, selected criteria related to war or refugees to discuss the comprehensive planning and reorganization of tourism space in the coastal areas of Syria. Zhou et al. [76] aimed at helping China to overcome poverty, and selected indicators related to household livelihoods to explore the evolution of the mechanism of remediation policies and measures among different stakeholders in the process of land governance. In the research on the spatial sustainability of agriculture [74,156] and forestry [153], some scholars have also considered the impact of household livelihood capacity.

#### 4.1.2. Measurement of the Index

Equations (1)–(3) were used to calculate the distribution of the sample literature index measurement method from the perspectives of time and topic, respectively. According to the meaning and dimensions of the index, it can be summarized in terms of Proportion, Quantity, Area, Distance, and Others.

##### Quantitative Analysis of Criteria

Figure 11 indicates several indices have shown a slightly fluctuating upward trend and have maintained a relatively stable ratio in the past five years. The ratio of the Proportion index is the highest, and the Distance index ranks the second. The Area and Quantity indices have obvious alternations in some years, and the overall number is comparable. 

Figure 12 shows that the ratio of Proportion is relatively high in Habitat and is the lowest in Site selection. The index of Quantity is the highest in Tourism, whereas it is lower in Site selection and Habitat. The Area index has the lowest overall proportion. The Distance index accounts for the largest proportion in Site selection, reaching 46%, whereas it is lower in other topics.

##### Content Analysis of Criteria

Table 7 shows the measurement content of the four types of indices in each topic. In addition to the above objective quantitative measurement methods, subjective and qualitative index measurement methods are also common. The use of a sociological questionnaire is one of the common ways to try to quantify the subjective wishes of respondents. In contrast to the Delphi method, the consultation objects are senior experts in the relevant professional fields. The objects of such questionnaires are usually the stakeholders. In Tourism, some scholars [132,139] conducted a questionnaire survey regarding the attractiveness, value, attention, and satisfaction of tourism resources, and finally obtained quantitative values of 1–10 or 0–1. In Habitat, some scholars [170,172] used a questionnaire to evaluate the accessibility, comfort, safety, and multivalences of space with scores in the range of 0–1. The qualitative analysis methods based on SWOT and TOWS are different from the common enrichment data collection process, and mainly construct a conditional descriptive comparison matrix [86,138] or a simple pairwise comparison matrix with no more than three levels [146] to measure the evaluation objectives.

### 4.2. Weight and Priority

Figure 13 shows that the number of studies using the subjective weighting method has grown steadily since 2011, and increased significantly since 2018, whereas the number of studies using the objective weighting method has gradually increased since 2017 and increased significantly in 2020. 

Figure 14 shows the distribution of all methods in different topics. Due to the characteristics of the AHP operating process, the combination of AHP and PCM is the most common in the topics of Site selection, Tourism, and Habitat. The number using subjective weighting methods is also significantly higher than that using objective weighting methods, and 34% of the studies used a comprehensive weighting method; only three studies used a combination of subjective and objective comprehensive weighting methods. Table 8 shows the classification of comprehensive weighting methods.

#### 4.2.1. Subjective Weight

Various types of subjective weighting methods are used in the sample literature. The weighting methods not only include the more traditional relative comparison methods, such as AHP, Delphi, PCM, and fuzzy approaches, but also the network methods represented by ANP and BWM. Combinations of multiple subjective weighting methods are common. Although Delphi is used in combination with PCM, the relative importance of factors in the pairwise comparison weight scale is determined by a group of experts related to the research topic. In Site selection, considering the practical significance of several level factors [105,110], AHP, ANP, fuzzy, Delphi, and DEMATEL methods have been used to assign weights to different factors at each level. 

Karasan et al. [107] used DEMATEL to identify the most effective criteria and their internal and external dependencies. Boyaci et al. [112] examined the problem of epidemiological hospital site selection in the context of COVID-19 and used the PFN-enhanced AHP method combined with GIS to determine the weight of criteria for the first time. Regarding Land use, Cui et al. [93], based on the combination of Delphi and PCM or AHP to determine the weights, introduced the fuzzy method to establish a semantic fuzzy judgment matrix, and finally used the fuzzy operator to comprehensively evaluate the degree of rural industry–city integration. Regarding Tourism, Zheng et al. [143] used Delphi by combining experts, field interviews, and website scoring to determine the weights among the factors in the directional ANP network of rural tourism transformation and development. Jeong et al. [11] proposed a comprehensive decision-making framework named MC-SDSS based on a web platform and several MCDA methods. Fuzzy-DAMATEL, PCM, and WLC were combined to ensure the relative objectivity and accuracy of weights.

#### 4.2.2. Objective Weight

The objective weights used in the sample are only used in Entropy, DEA, and BP networks. Regarding Land use, some scholars [77,85,87] introduced the Entropy method mainly to revise the weights determined by existing research or the mathematical models used. In Tourism, researchers [132,136] used the Entropy method to assign weights to the spatial factors affecting the accessibility and attractiveness of tourist destinations. In Urban–rural planning, some researchers used it to determine the weight of traffic accessibility indicators in rural mountainous areas [113,118], the spatial factors affecting eco-agricultural development [115], or the spatial factors that influence the orientation of landscape functions [127]. From the perspective of the use of cross-step methods, the combination of Entropy as a weighting method and TOPSIS as a ranking method is the most commonly used approach [79,80,113,118,127,164]. In addition, as an existing linear programming model, the DEA method does not involve artificial weight distribution or calculations in the operation process, and is mainly used in the evaluation of tourism industry productivity [137], leisure agriculture efficiency and driving factor evaluation [141], and the evaluation of the efficiency of tourism industry poverty alleviation [133]. According to the advantages of the BP network in dealing with uncertain nonlinear complex relationships, Kong et al. [123] proposed an improved PSO-BP network, which modifies the weights of neurons in each layer through machine learning to reduce the error signal, to determine the sorting potential of rural settlements in poor mountainous areas by continuous learning and tuning.

#### 4.2.3. Comprehensive Weight

The number of studies using the comprehensive weight method combining subjective and objective weights is still insufficient. Zhou et al. [78] used Entropy combined with Delphi and PCM to determine the rate of contribution of government support, environmental pressure, resource endowment, and quality of life to the spatial differences in rural eco-development in arid regions of China. In Urban–rural planning, scholars [114,119] combined Entropy with Delphi and AHP, and finally took the average of expert scores and entropy values as the weight of multiple factors affecting the quality or potential of sustainable development in villages and towns.

### 4.3. Ranking and Verification

#### 4.3.1. Ranking and Aggregation

##### Quantitative Analysis of Ranking and Aggregation

Figure 15 and Figure 16 show that AHP remains the most commonly used ranking method, both in total and in each research topic, and its use has continued to rise over the past five years. After AHP, TOPSIS and ANP are the second and third most frequently used, respectively. TOPSIS is the most widely used method, after AHP, in Site selection and Urban–rural planning. ANP is the most popular method in Habitat.

Aggregation is the final step to determine alternatives after sorting, and can be divided into the Hard Mathematical method, Soft Mathematical method, and Voting method, according to [31]. Table 9 shows the usage frequency of all aggregation methods. The usage frequency of the Hard Mathematical method is 75.73%, and this is the most commonly used aggregation method. Both the Soft Mathematical method and Voting method are at lower levels, accounting for 14.48% and 3.88%, respectively.

##### Content Analysis of Ranking and Aggregation

The TOPSIS method in Site selection or Urban–rural planning is widely used to deal with site suitability [9,98,102,103,107,112], scheme comparison [115,118], traffic accessibility [113,118], and influencing factors [79,80] by calculating the Euclidean distance between alternative projects and the optimal and worst solutions. ANP is used to solve system problems with a feedback mechanism in villages [116,143,162,169]. The fuzzy method is used to solve the dilemma that a clear mathematical model cannot be established in the whole process of MCDA or in a certain link, so is often combined with other MCDA methods [75,102,104,106]. DEA is used to assess the characteristics or influencing factors of tourism development by calculating the ratio between inputs and outputs [133,137,141].

In the aggregation calculation stage after ranking, the Hard Mathematical method selects the optimal result by calculating the sum and average of the values (or scores) of all or specific alternatives. Based on the Hard Mathematical method, the Soft Mathematical method adds the human intervention of decision makers, and is used to solve the research problems of public participation [108,128,146,147] and conflicts of views [86,126,138,152]. The voting method determines the optimal result by selecting a number of stakeholders involved in the research problem to vote on the alternatives, and is mainly used to solve research problems of public participation and community co-governance [93,122,145,157].

#### 4.3.2. Verification of Data

##### Quantitative Analysis of Verification

According to the review of the sample, the verification methods can be divided into Empirical data, Previous research, Comparison between methods, SA, Monte Carlo, and some combinations of these. Figure 17 shows the distribution of all verification methods in different topics. Due to forecasts or lack of data, most of the studies did not verify the results. The usage rate of Empirical data ranked second; SA was used more frequently in Land use and Site selection.

##### Content Analysis of Verification

As one of the more commonly used verification methods, SA is used 11 times in the sample [11,82,91,92,94,95,100,109,110,112,158], and is mainly used to test the standard score (or performance measure) of the effectiveness of alternative ranking. Empirical data are verified by comparing the results of MCDA with the follow-up data of existing research objects or the existing situation of the implemented program. This approach is widely used to deal with regularity [86,90,120,155], the mechanism [87,149,156], and suitability [84,96,98,99,103,106] in Conservation, Land use, and Site selection. The comparison between methods achieves the purpose of testing by comparing the results of multiple MCDA methods [85,104,109]. The operation principle of previous research is based on the data or results of similar previous research as a reference for data verification [89,167].

## 5. Discussion

### 5.1. Discussion of the Results in Primary Review Analysis

#### 5.1.1. Overview of Research Trends

The results in Section 3.1 show that 2015 and 2018 were the years in which the related research increased rapidly. In 2015, with the signing of the Paris Agreement, a unified global system of sustainability goals was established. Then, in 2018, the Central Committee of the Communist Party of China incorporated the concept of green development into the new rural construction plan [174], which further promoted the attention of Chinese academia on the spatial sustainability of rural areas. Driven by international events and national policies, the academic community began to reach a consensus [175,176], realizing that consideration of several factors and systems in rural areas is an important way to promote sustainable development.

The results in Section 3.2 show that the research areas are mainly concentrated in developing countries or developing regions in developed countries. Because urban and rural development and its sustainability in developed countries have entered the era of the inventory update [177], and developing countries are still in the process of continuous urban and rural construction [178,179], global scientific research institutions and universities are paying more attention to the spatial sustainability research in rural areas.

#### 5.1.2. Research Topic and Unit Range

The results in Section 3.3 show that the literature can be divided into six research topics. It can also be seen that related research has been widely focused on Environmental Sciences and Ecology, and the combination with the GIS platform has matured in the past 20 years via the use of clustering analysis [180]. From the results in Table 7, we can further see that existing studies are closely related to society, the economy, and the environment in the six topics. However, culture, as mentioned in “Culture: at the heart of SDGs”, proposed by the *United Nations Educational**, Scientific, and Cultural Organization* [181], only involves Urban–rural planning, Tourism, and Habitat. The number of relevant studies is still insufficient. Under the influence of frequent urban renewal, geopolitics, and boundary replacement, the preservation of cultural heritage is as important as the protection of the ecological environment.

The results in Section 3.4 show that a large amount of existing research is focused on the macroscopic scale. The number of studies regarding Community/village is lower, and studies of Architecture are particularly scarce in the existing research, compared to the large-scale data acquisition that relies on satellites, remote sensing, monitoring, and even access to existing databases and data platforms. Extensive reliance on field surveys greatly increases the difficulty of data acquisition in small- and medium-scale research. The lack of specific data may have contributed to the scarcity of relevant studies.

The results in Section 3.5 show that the sample is dominated by traditional quantitative methods for enriching data. As shown by the frequency ranking of the MCDA method shown in Figure 8, the simple MCDA method is used often in each research topic [31,73]. Most of the studies using AHP in the weighting stage still continue to use it in the ranking step because of its versatility. Due to the convenience of mathematical model establishment and the existence of current software (such as Yaahp, EvaGear, and Super Decisions), Delphi, PCM, TOPSIS, and ANP are more frequently used. In recent years, the qualitative MCDA method has also begun to emerge gradually due to its simple operation and convenience; however, the number, depth, and accuracy of related studies are still insufficient.

### 5.2. Discussion of the Results in Detail Review Analysis

#### 5.2.1. Criteria Aspects

The results in Section 4.1.1 show that the number of environmental criteria dominates, whereas social criteria contribute more to Habitat, and economic criteria need to be considered in Land use and Urban–rural planning. In the context of the current global environmental crisis, from the multi-dimensional perspective of elements and characteristics, focusing on a comprehensive evaluation of the status quo and potential of the environment has become an overall trend in the research on the sustainability of rural spaces around the world. 

The results in Section 4.1.2 show that Proportion is the most commonly used index. Since most of the statistical data in yearbooks and government websites are presented in the form of proportions, the convenient access channels have led to the dominance of the Proportion index. In Site selection, measuring the distance between candidate addresses and the sensitive locations of the research question has been verified as the most effective approach by many studies [97,103,110]. However, compared to using straight-line distance measurement, in many cases [9,20,103,104,107,110,111] using distance to measure may be more rigorous and scientific. The insufficient application of the Area index in the evaluation of rural spatial sustainability and few measurable factors are the weak points of the current research.

#### 5.2.2. Feature and Distribution of MCDA Methods

The results in Section 4.2 show that the combined use of subjective weighting methods has dominated during the past 10 years due to the superiority of subjective weighting over objective weighting in terms of operational convenience. On this basis, PCM and Delphi are widely used for their simplicity and compatibility. The practical significance of criteria is considered in the formulation process in subjective weighting, but is greatly affected by the knowledge background and the existing cognitive structure of the researcher establishing the criteria. The objective weighting eliminates human interference, but does not consider the actual meaning of the indicator representation. Few existing studies have used the combination of subjective and objective weights, but such a practice is very common in the fields of computer and management science [182,183]. The results in Section 4.2.2 shows that 12 of the 14 studies using objective weights were conducted by scholars in the Chinese region, and only two are from other countries. Chinese scholars and their related research tend to use the method of objective weighting. At present, no existing literature or data can explain the reasons for this phenomenon. According to the experience of the research team working in the countryside for many years, the complexity and contradiction of multiple stakeholders in rural China is one of the possible reasons for the unconvincing use of subjective weighting.

The results in Section 4.3 shows that AHP and TOPSIS are still the mainstream ranking methods; this finding is also in line with Gebre et al. [27]. Secondly, the effective application of TOPSIS to Site selection and Urban–rural planning has been demonstrated. The Hard Mathematical method is a widely used data aggregation method. The Soft Mathematical and Voting methods provide a more democratic aggregation process for public research problems that may cause multi-stakeholder conflicts [31]. The existing research generally lacks a verification process. It is common to compare and verify the MCDA evaluation results through the existing empirical data, whereas the evaluation research of the prediction type is often unable to verify the results due to the lack of effective follow-up data. In this case, mathematical verification models such as SA and Monte Carlo may help to verify the validity of the results.

### 5.3. Limitations

The literature reviewed in this study is limited to the WOS, Scopus, and CNKI databases. Because the bibliometric analysis of the database generated by only relying on a string search may produce misleading results, the manual screening process of the team was added on the basis of an automatic search. Although the researchers tried their best to screen out the literature that contains all relevant keywords, some literature may have been missed due to some unique keywords. Therefore, establishing a literature capture or cleaning method based on advanced algorithms is an important technical path to more accurately determine the relevant literature in the future.

In this review, a detailed analysis of other software or tools used in conjunction with the MCDA method was not performed. The combination of the MCDA method with interdisciplinary software or tools has implications for improving the accuracy of rural spatial assessments. Therefore, its specific coupling usage rules need to be further explored in future research work.

## 6. Conclusions

Based on the overall trends and research content, this study conducted a comprehensive review of 103 studies around the world according to the three common operation steps of the MCDA method and vertical and horizontal comparisons.

(1)During the past 5–8 years, the evaluation of spatial sustainability in rural areas has gradually become a popular research topic due to the worldwide environmental crisis. More research has been published in the Eastern Hemisphere, less in the Western Hemisphere, more in the Northern Hemisphere, and less in the Southern Hemisphere. China, Turkey, and Italy are the three countries producing the most publications. Developing countries or developing regions in developed countries comprise an important zone for related research.(2)The literature can be divided into six research topics: Land use, Site selection, Urban–rural planning, Tourism, Conservation, and Habitat. The target area of the study consists of four scales: Regional scale, District/county area, Community/village, and Architecture. Macro-scale research is relatively mature, but that of Communities/villages and Architecture is still insufficient.(3)Due to the urgent needs of the environment, the impact of environmental indicators on research problems and goals should be first considered in the process of criteria formulation. Social and economic criteria are secondary considerations. The contents of other criteria can be formulated with reference to Table 6. Proportion is the most commonly used index type, and Distance is one of the most effective index types in Site selection. The content involved in the index can be formulated with reference to Table 7.(4)The subjective weighting method is still used in mainstream research, with AHP being the most prominently used approach. Delphi and PCM have excellent applicability in most research topics. For the problems in Site selection and Urban–rural planning, TOPSIS is a more effective MCDA method. Objective weighting is an ideal method for solving problems in Land use, Urban–rural planning, and Tourism. The evaluation of rural spatial sustainability is a very complex problem, and the use of only a single MCDA method has limitations. A hybrid MCDA approach is the preferred option. The Hard Mathematical method can handle most situations, but the Soft Mathematical and Voting methods are recommended for issues involving public participation. The comparative verification of empirical data is a more intuitive method, but when data are lacking, statistical models can be considered.

Based on the above analysis, the following points were identified for future research work:(1)This study found that, although there are fewer rural studies in developed countries, they are limited by the method and region. Hence, determining the difference in the focus of research on spatial sustainability assessment in developed and developing countries is one of the issues worthy of further study.(2)In the current research on the evaluation of the sustainability of rural space, the consideration of cultural factors is one of the research gaps. As the core of sustainable development goals, cultural factors need to be paid more attention. Therefore, the research of sustainable space evaluation related to rural cultural heritage will be an important topic for future work.(3)Against the background of a rapid marginal expansion of related research fields and frequent interdisciplinary penetration, establishing a relatively complete data collection and sharing mechanism within the scope of small- and medium-scale units is basic work that urgently needs to be undertaken in the research on rural spatial sustainability.(4)Important means to improve the accuracy of rural spatial sustainability evaluation include expanding the types of feasible measurement criteria and indices; increasing the combination of subjective and objective weights; further clarifying the boundary conditions for the use of the deterministic MCDA methods; and adding auxiliary verification for predictive evaluation.(5)Identifying different characteristics and orientations of rural spatial sustainability research issues, and adopting appropriate quantitative or qualitative MCDA methods, can provide a theoretical basis for formulating a scientific and reasonable evaluation system, so as to provide a decision-making reference for more active and effective promotion of rural sustainable development. The proposal of a scientific and comprehensive MCDA selection list according to research fields, topics, and goals is still an important challenge for future research.

This review contains a wealth of up-to-date literature data and information on the application of multi-criteria decision-making analysis to the problems of evaluating rural spatial sustainability. It shows the current state of MCDA in related fields. The resulting report can serve as a guide for scholars, researchers, policymakers, and, in particular, natural resource managers, to establish and improve decision-making techniques, as a basis for further research, and to select appropriate decision-making techniques to develop a framework for spatial evaluation.

## Figures and Tables

**Figure 1 ijerph-19-06572-f001:**
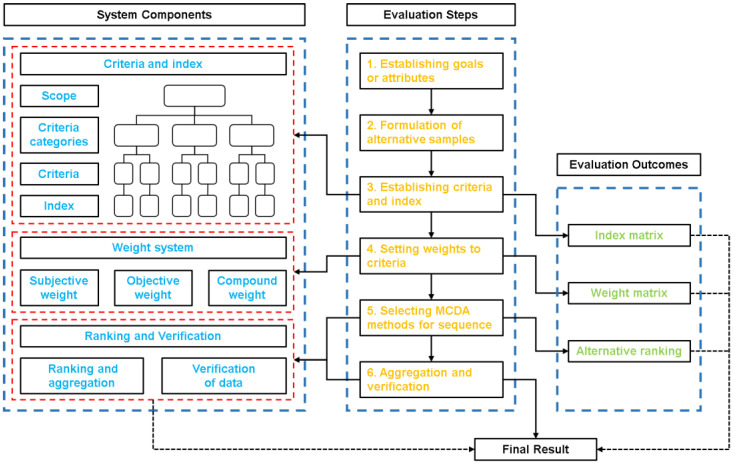
Operation process and system composition of MCDA.

**Figure 2 ijerph-19-06572-f002:**
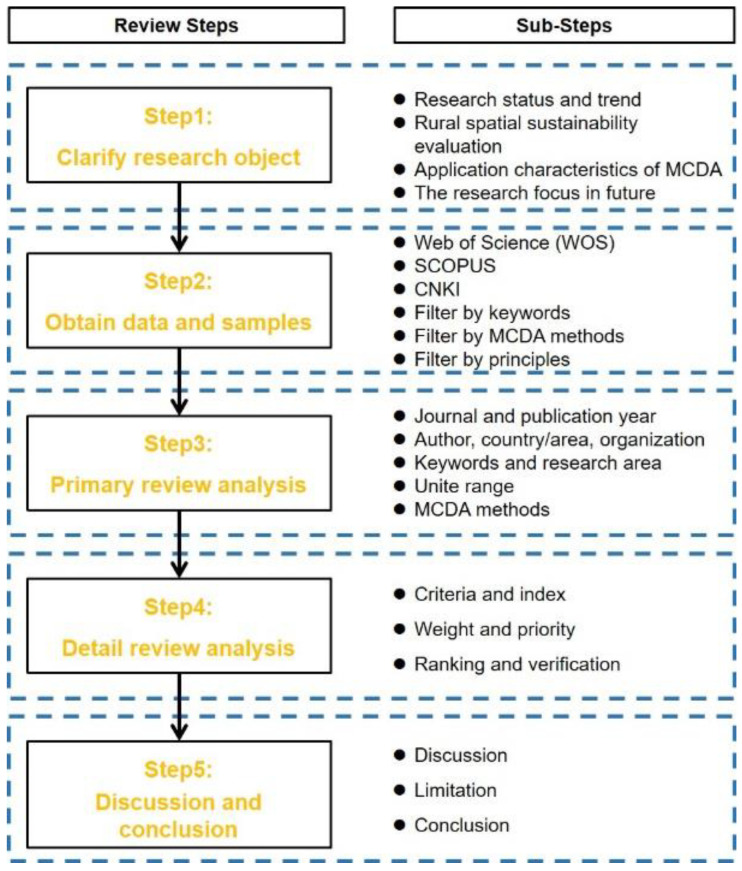
The structure of the research.

**Figure 3 ijerph-19-06572-f003:**
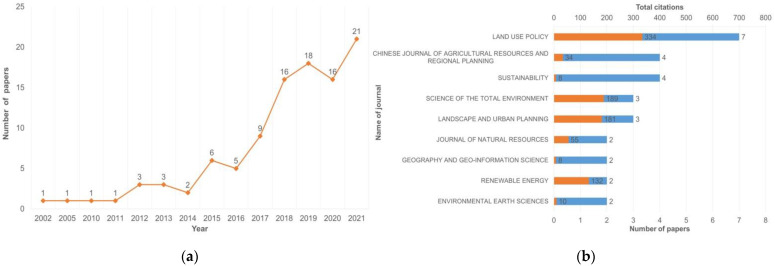
(**a**) Trend of published articles based on publication year; (**b**) distribution of journal publications and citations. The orange bar represents the number of citations, and the blue bar represents the number of publications.

**Figure 4 ijerph-19-06572-f004:**
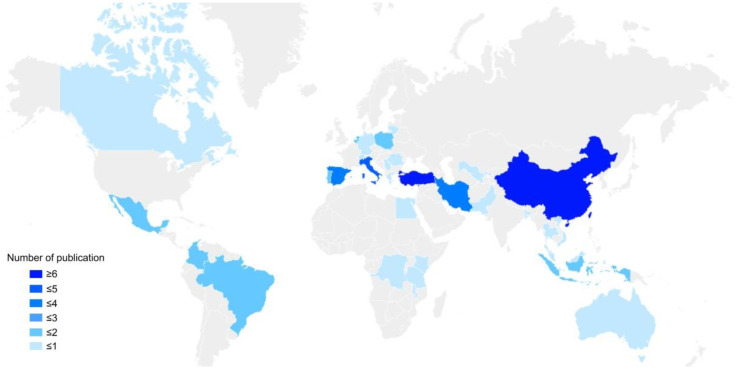
Distribution of publications based on country/area.

**Figure 5 ijerph-19-06572-f005:**
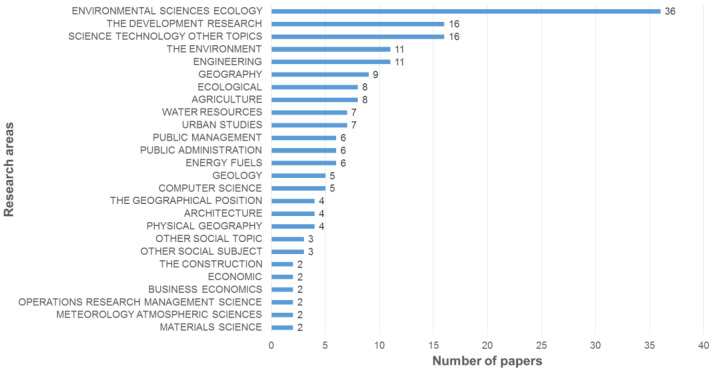
Distribution of publications based on research areas.

**Figure 6 ijerph-19-06572-f006:**
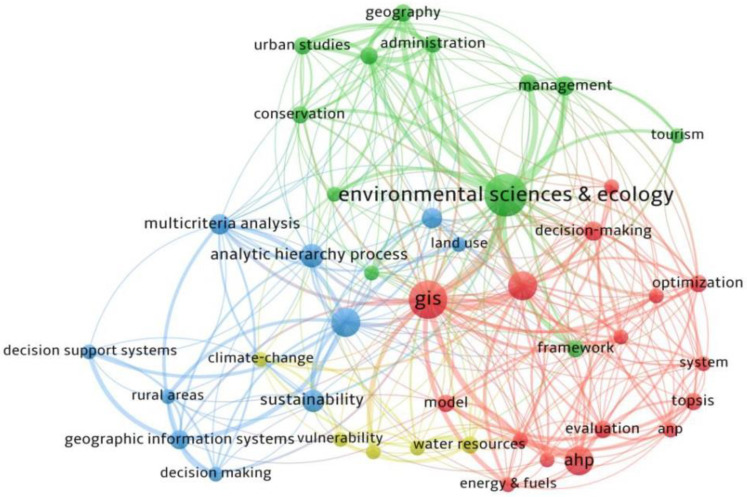
Keywords occurrence and clustering by VOSviewer 1.6.17.

**Figure 7 ijerph-19-06572-f007:**
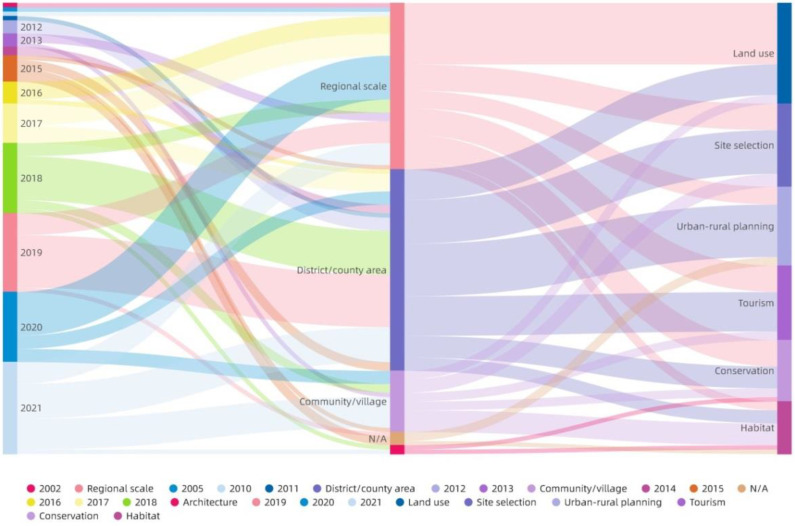
Sankey diagram of year, unit range, and research topic distribution.

**Figure 8 ijerph-19-06572-f008:**
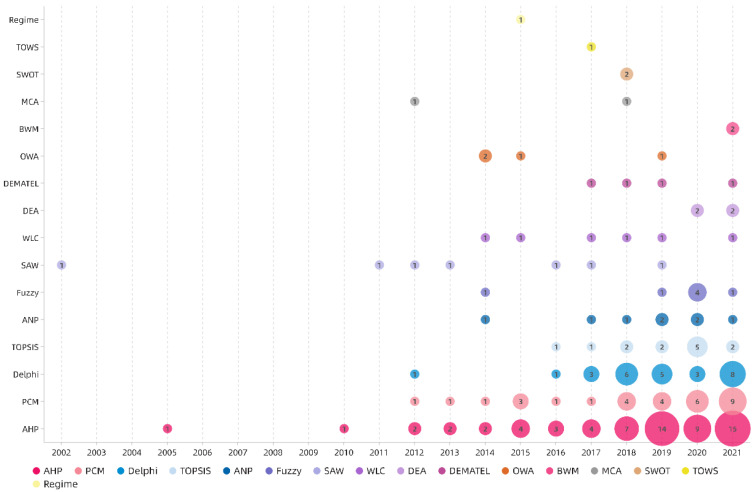
The annual distribution of occurrences of the MCDA methods. The size of the bubble and the number in the bubble indicate the number of times the method was used in the corresponding year.

**Figure 9 ijerph-19-06572-f009:**
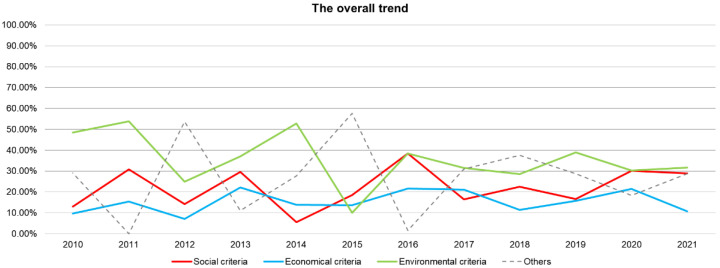
The annual distribution of the proportion of criteria categories.

**Figure 10 ijerph-19-06572-f010:**
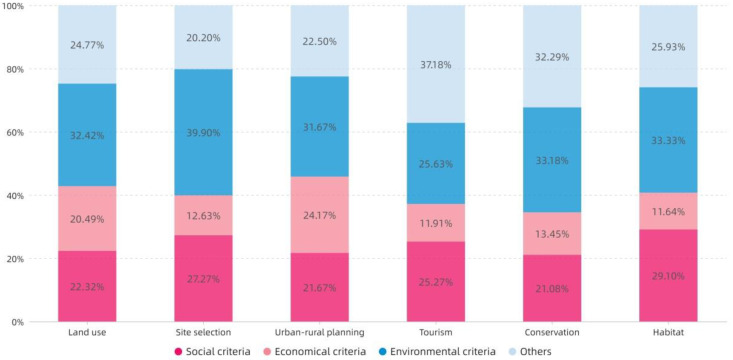
The topic distribution of the proportion of criteria.

**Figure 11 ijerph-19-06572-f011:**
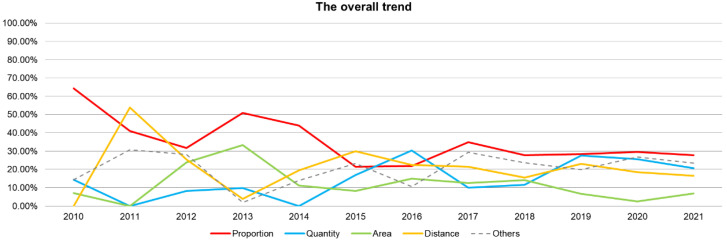
The annual distribution of the proportion of index categories.

**Figure 12 ijerph-19-06572-f012:**
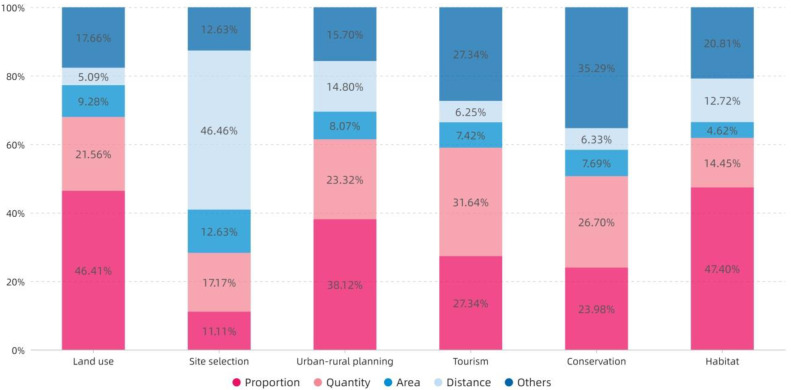
The topic distribution of the proportion of index categories.

**Figure 13 ijerph-19-06572-f013:**
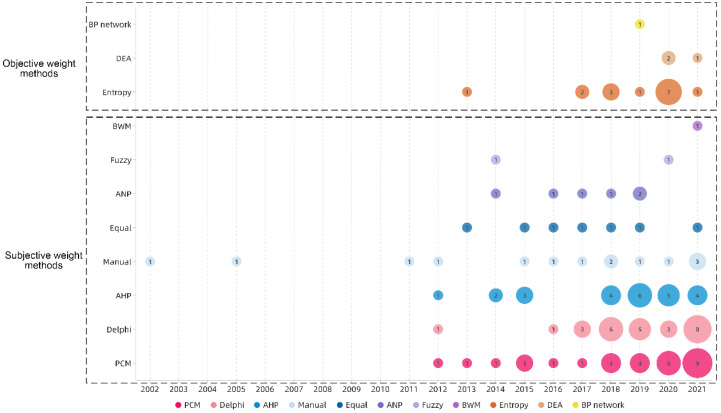
The annual distribution of occurrences of the weighting methods.

**Figure 14 ijerph-19-06572-f014:**
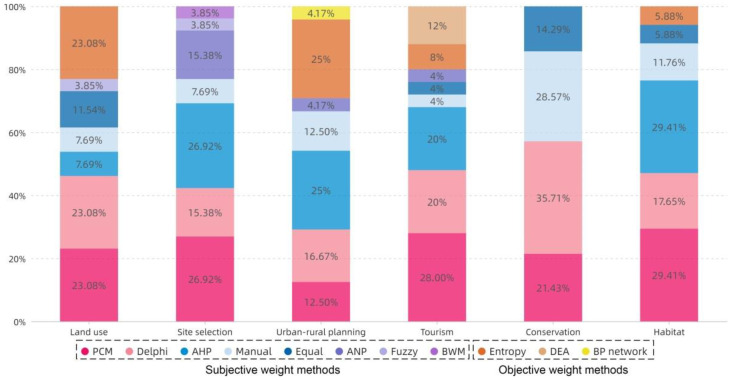
The topic distribution of the proportion of weighting methods.

**Figure 15 ijerph-19-06572-f015:**
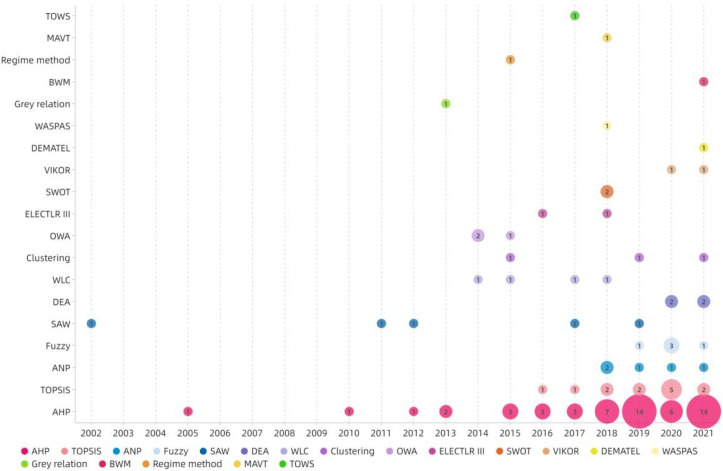
The annual distribution of occurrences of the ranking methods.

**Figure 16 ijerph-19-06572-f016:**
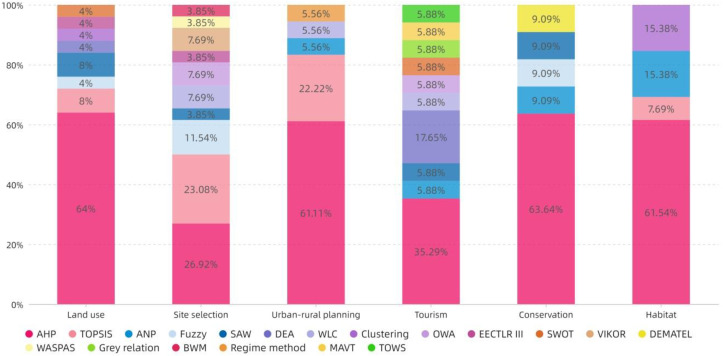
The topic distribution of the proportion of ranking methods.

**Figure 17 ijerph-19-06572-f017:**
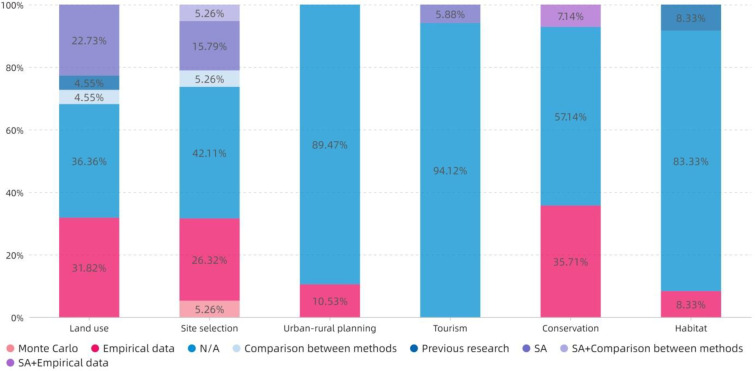
The topic distribution of the proportion of verification methods.

**Table 1 ijerph-19-06572-t001:** Summary of existing related reviews based on the MCDA method.

Ref.	Number of Documents	Time Period	MCDA Types	Analysis Method	Research Focus	Findings
[27]	69	2000–2019	26 (including compound methods)	Cluster analysis	Rural land resource allocation	Status and trends
[28]	46 (MCDA apllied)/84	2000–2020	10	Cluster analysis	Location of flood hazard/susceptible regions	Observations and recommendations

**Table 2 ijerph-19-06572-t002:** Classification of MCDA methods based on functional perspective.

Method	Function	Introduction	Ref.
AHP	Subjective weight ranking	AHP was first developed by Saaty (1980). It consists of a pairwise comparison, using relative values of criteria (weights) and a score scale on alternatives against criteria. Typically, eigenvalue techniques or regression analysis are used to determine the ordering. AHP is one of the most common MCDA methods.	[36]
ANP	Subjective weight ranking	The analysis process of ANP and AHP is roughly the same, but in contrast to the one-way process of AHP, ANP has a feedback mechanism.	[37]
BP network	Objective weight ranking	BP network is a type of supervised network based on error back-propagation. Through the calculation and feedback of input information, output information, and the error, the weights of neurons in each layer of the BP network can be modified to obtain the minimum error signal and form the final network model. The learning process of the neural network is mainly the updating process of weights.	[38]
BWM	Subjective weight ranking	BWM can obtain weights for different criteria based on pairwise comparisons, requiring less comparison data. Quite differently from AHP, BWM only performs reference comparisons, which means it only needs to use numbers between 1 and 9 to determine the preference of the best criterion over all other criteria and the preference of all criteria over the worst criterion. This process is easier, more accurate, and less redundant because it does not perform quadratic comparisons.	[16,39,40]
DEA	Objective weight ranking	DEA is a nonparametric productivity measure for multiple-input and multiple-output operations. The method combines and converts multiple inputs and outputs into a single efficiency metric. This approach first establishes an “efficient frontier” formed by a set of decision-making units (DMUs) exhibiting best practices, and then assigns efficiency levels to other non-boundary units based on their distances to the efficient frontier. Finally, the efficiency level of each object is obtained.	[41]
Delphi	Objective weight	The Delphi method is a systematic, interactive method that relies on a group of independent experts. Based on this principle, Delphi uses carefully selected experts to answer questionnaires to determine weights. After each round, the summary of the previous round of expert selection and the reasons for their judgment will be fed back to the experts. During this process, the floating range of the weights will shrink and gradually converge towards the “correct” weights. Finally, the process is stopped according to predefined rules.	[42]
DEMATEL	Ranking	DEMATEL is considered for an effective method for identifying the components of the causal chain of complex systems. It handles the interdependencies among the evaluation factors and finds the key factors through a visual structural model. The DEMATEL technique can convert the interrelationships between factors into an understandable structural model of the system and divide it into cause and effect groups. Its operation consists of four steps: 1. establish an influence matrix; 2. create a normalized matrix; 3. build a total influence matrix; 4. generate an influence graph.	[43]
ELECTRE	Ranking	ELECTRE is a technique for selecting the best alternative from a given set of alternatives. It uses pairwise comparisons of alternatives and sets ranking relationships on them. For example, if a is at least as good as b for criterion i, then alternative a is ranked higher than the alternative b. The ELECTRE tool has evolved over time into different sequential versions, such as ELECTRE I, II, III, IV, and ELECTRE TRI.	[44,45]
Entropy	Objective weight	Information entropy is a measure of the degree of disorder in a system. It can measure the amount of useful information with the data provided. When the value difference between the evaluation objects of the same indicator is large, but the entropy is small, it means that the indicator provides more useful information, and the weight of the indicator should be set correspondingly high.	[46]
Fuzzy	Subjective weight Ranking	Due to the availability and uncertainty of information, as well as the ambiguity of human perception and cognition, most selection parameters cannot be given accurately. The fuzzy set provides a mathematical model to determine the membership degree of each element to the set, so that the applicability evaluation data of various subjective standards and the weight of the standard are converted into numerical values by the language of the decision maker.	[47]
Grey relation	Ranking	The grey relational method is a branch of grey system theory developed in 1980. The method is similar to TOPSIS, which defines the grey relational degree to represent the closeness between the alternatives. Typically, ideal scenarios are defined and the degree of relevance of alternatives to them is calculated. The most relevant alternative has the shortest distance from the ideal solution and the longest distance from the worst solution.	[48]
MAVT (MAUT)	Ranking	MAUT is a systematic approach that takes into account decision makers’ preferences in the form of a utility function defined over a set of attributes. Its functional form is determined by applying preference validation and setting certain utility-independent conditions. The formula is extended to derive a multi-attribute utility function. In the case of MAVT, the difference between utility and value is that the function uses attribute values instead of quantitative attribute measures in a cardinal scale.	[49]
OWA	Ranking	The OWA method is similar to WLC, but considers two sets of weights. The first set of weights controls the relative contributions of specific criteria, and the second set of weights controls the order in which the weighted criteria are aggregated. The advantage of OWA is that a variety of different solutions and forecast scenarios can be generated by reordering and changing standard parameters.	[50,51]
PCA	Objective weight	The modeling properties of PCA are largely rooted in regression thinking: variation explained by principal components. After introducing the idea of linear combination of variables, the change in principal components is emphasized. When there is a certain correlation between the two variables, it can be explained that the information of the two variables reflecting the subject overlaps to a certain extent. Principal component analysis is to delete duplicate or irrelevant variables for all the variables originally proposed, and establish as few new variables as possible, so that these new variables are unrelated to each other. Finally, the importance of the variable is obtained.	[52]
PCM	Subjective weight	In the pairwise comparison method, participants are presented with a worksheet and asked to compare the importance of two criteria at a time. The scoring scale can be varied, for example, an odd scale of 1 to 9 is often used. Results are combined by adding the scores obtained for each criterion, when the preferred criterion is compared with the criterion to which it is compared. The results are then normalized to a total of 1.0. This weighting method provides a framework for comparing each criterion to all others and helps to show differences in importance between criteria.	[31]
Regime	Ranking	The regime method is a discrete multiple evaluation method suitable for evaluating projects and policies by processing qualitative and quantitative information. It uses pairwise comparisons to evaluate the performance of alternatives and establishes ranking relationships among alternatives. The framework of the method is based on two kinds of input data: an influence matrix and a set of weights.	[53,54]
SAW	Ranking	The SAW method, originally applied by Charles (1954), is one of the most commonly used MCDM techniques. The method performs a simple multi-product summation of each criterion score through the corresponding attribute weights to find an overall performance measure for each alternative.	[55]
SWOT	Ranking	SWOT analysis is a common tool for strategic planning and a form of brainstorming. It helps organizations better understand their internal and external business environment when making strategic plans and decisions by analyzing and locating their resources and environments in terms of four areas: strengths, weaknesses, opportunities, and threats.	[56,57]
TOPSIS	Ranking	TOPSIS means that the optimal selection scheme has the shortest Euclidean distance from the ideal solution and the largest distance from the negative ideal solution. Intuitively, based on the distance from the ideal solution, the method can take any number of attributes as input. However, TOPSIS can produce unreliable results, and it also does not account for the uncertainty of the weights.	[58]
TOWS	Ranking	TOWS matrix is a derivative type of SWOT. In contrast to the SWOT method, TOWS focuses more on the solution strategy obtained through the situation analysis. The matrix includes four strategies: WT, WO, ST, and SO.	[59]
VIKOR	Ranking	The VIKOR method is similar to the TOPSIS method in that both are based on distance measurements. In contrast to the strict sorting of TOPSIS, VIKOR seeks a compromise solution. The VIKOR method can also provide clustering capabilities when faced with alternatives.	[60,61]
WASPAS	Ranking	The WASPAS method combines the historical data and current data, and adds the weighted sum model (WSM) and the weighted product model (WPM) to determine the decision target under the corresponding decision criterion.	[62]
WLC	Ranking	WLC is an evaluation function method, and is a method of solving multi-objective/attribute programming problems by assigning corresponding weight coefficients to each objective according to its importance, and then optimizing its linear combination.	[63]

**Table 3 ijerph-19-06572-t003:** Bibliographic database source characteristics.

Type	Feature	Search Result	Strength	Weakness	Publisher	Ref.
Web of Science (WOS)	An interdisciplinary platform with many scientific databases	Advanced search function Searches are reproducible and reportable Reliable screening	Search function with wide selectable categories	Moderate coverage of interdisciplinary journals	Clarivate Analytics	[65]
Scopus	Natural Sciences, Engineering, Social Sciences, Biomedical Sciences, Arts and Humanities	Advanced search function Searches are reproducible and reportable Reliable screening	Search function with wide selectable categories	Difficult to obtain the full text of some documents	Elsevier	[27]
Science Direct (SD)	Natural Science, Technology and Medicine	Normal search function Reliable screening	Easy to search articles by journal	Higher data repeatability with Scopus	Elsevier	[27]
Google Scholar	All subject areas	Simple search function No reproduce and report	Contains almost all types of files	Few sorting options Numerous Non-peer-reviewed data sources	Google	[66]
China National Knowledge Infrastructure (CNKI)	An interdisciplinary platform with many scientific databases	Advanced search function Searches are reproducible and reportable Reliable screening	Search function with wide selectable categories	Chinese literature dominates	Tsinghua University/Tongfang Co., Ltd.	[67]

**Table 4 ijerph-19-06572-t004:** Summary of important authors and institutions in related fields.

Author (* Is the First Author)	Institution	Times of Citations	Number of Papers	Publication Year (Citation of Article)
Jin SuJeong *	University of Merida, Spain	137	4	2012 (46), 2014 (55), 2015 (16), 2018 (20)
LorenzoGarcía-Moruno	University of Merida, Spain	91	3	2014 (55), 2014 (26), 2015 (16)
JulioHernández-Blanco	Universidad de Extremadura, Spain	71	2	2014 (55), 2015 (16)
Xu, Xiaodong	Southeast University, China	8	2	2018 (3), 2019 (5)
Ren, Guoping *	China Agricultural University, China	25	2	2018 (25), 2021 (0)
Liu, Liming	China Agricultural University, China	25	2	2018 (25), 2021 (1)

**Table 5 ijerph-19-06572-t005:** Summary of research topics.

No.	Research Topics	Number	Ref.
1	Land use	22	[74,75,76,77,78,79,80,81,82,83,84,85,86,87,88,89,90,91,92,93,94,95]
2	Site selection	19	[9,20,96,97,98,99,100,101,102,103,104,105,106,107,108,109,110,111,112]
3	Urban–rural planning	19	[113,114,115,116,117,118,119,120,121,122,123,124,125,126,127,128,129,130,131]
4	Tourism	17	[11,132,133,134,135,136,137,138,139,140,141,142,143,144,145,146,147]
5	Conservation	14	[148,149,150,151,152,153,154,155,156,157,158,159,160,161]
6	Habitat	12	[162,163,164,165,166,167,168,169,170,171,172,173]

**Table 6 ijerph-19-06572-t006:** Summary of criteria other than economy, society, and environment.

No.	Research Topics	The Content of the Other Criteria
1	Land use	Aspect, Slope, Soil fertility, Precipitation
		[75,82,83,87]
2	Site selection	Geology Professional Indicators, Technology and Policy Elements
		[107,110]
3	Urban–rural planning	Culture, Form, Function, Location Accessibility
		[123,125,126,128,131]
4	Tourism	Aesthetic effect, Cultural heritage, Agricultural entertainment, Information construction
		[135,136,139,141,143,146,147]
5	Conservation	Management ability, Physical basis, Technology and Policy
		[148,152,154,158,161]
6	Habitat	Culture, Form, Function, Location Accessibility
		[165,172]

**Table 7 ijerph-19-06572-t007:** Summary of index content.

	Proportion	Quantity	Area	Distance
Land use	Funding, Time, Correlation, Covering, Population, Resource, Development	Appliance, Facility, Production, Time, Pollution, Funding, Resource, Training Committee	Farm, Location, Water, Soil	Road, Facility, Location, Transportation hub, Water, Forest, Grid, City
	[78,82,84,85,90,93,95]	[77,78,80,84,85]	[79,81,92]	[75,82,87]
Site selection	Covering, Development	Funding, Time, Resource, Disaster, Development	Farm, Location, Zone, Water, Soil	Community, Road, Facility, Location, Transportation hub, Water, Forest, Grid
	[110]	[107]	[98,99,101,108]	[20,101,103,110]
Urban–rural planning	Funding, Correlation, Covering, Building, Population, Development	Appliance, Facility, Location, Building, Production, Pollution, Funding, Resource,	Farm, Location, Zone, Water, Soil, Landscape	Road, Facility, Location, Transportation hub, City, County
	[114,115,119,123,127]	[114,121,127]	[127,128]	[113,119,123,128]
Tourism	Funding, Covering, Building, Population, Development	Facility, Location, Building, City, County,	Location, Zone, Water, Soil, Landscape	Road, Facility, Location, City, County
	[134,139,142]	[132,134,139,140,142]	[11,140,144]	[140,144]
Conservation	Funding, Covering, Population, Resource, Development	Appliance, Facility, Location, Building, Culture, Funding, Committee	Resource, Location, Zone, Water, Soil, Landscape	Road, Facility, Location, Transportation hub, Water, Forest,
	[154,160,161]	[148,154]	[158,160]	[158,159]
Habitat	Pollution, Population, Production, Covering, Building, Road, Resource, Space, Development	Appliance, Facility, Location, Building, Culture	Zone, Landscape	Road, Facility, Location, City, Transportation hub, Water, Forest,
	[164,167,169,171]	[167,170]	[172]	[163,172,173]

**Table 8 ijerph-19-06572-t008:** Summary of comprehensive weighting methods.

Types	Methods	Ref.
Subjective and Objective Weight	Delphi + AHP + Entropy	[114]
	Delphi + PCM + Entropy	[78]
	Entropy + AHP	[119]
Multiple Subjective Weights	ANP + AHP + Fuzzy	[105]
	AHP + PCM	[98,100,102,108,125,129,135,140,144,145,165,169,172]
	Delphi + AHP	[91,121,163]
	Delphi + ANP	[143]
	Delphi + DEMATEL + ANP	[110]
	Delphi + Fuzzy + AHP	[93]
	Delphi + PCM	[95,106,134,139,150,162,170]
	Delphi + PCM + AHP	[142]
	DEMATEL + ANP	[107]
	PFS + AHP	[112]
	MC − SDSS	[11]

**Table 9 ijerph-19-06572-t009:** Summary of the proportion of aggregation methods.

NO.	Aggregation Method	Ratio	Ref.
1	Hard Mathematical method	75.73%	[9,11,20,74,75,76,77,78,79,80,81,82,83,84,85,87,88,89,90,91,92,94,95,96,97,98,99,100,102,103,104,106,109,110,111,112,113,114,115,116,118,119,120,121,123,125,127,129,130,132,133,134,135,136,137,139,140,141,142,143,144,153,154,155,156,158,159,160,162,163,164,166,167,169,170,171,172,173]
2	Soft Mathematical method	14.48%	[86,101,105,107,108,124,126,128,131,138,146,147,148,149,150,151,152,161]
3	Voting method	3.88%	[93,122,145,157]
4	N/A	2.91%	[117,165,168]

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
