# Peer review of "Application of Multi-Criteria Decision-Making Analysis to Rural Spatial Sustainability Evaluation: A Systematic Review"

_ijerph, 2022, doi:10.3390/ijerph19116572_

Round 1

Reviewer 1 Report

This paper thoroughly reviewed the previous researches of rural spatial sustainability evaluation using multi-criteria decision-making analysis. The review is comprehensive and systematic. There are some minor problems need to be fixed.

Tittle

Framework construction of rural spatial sustainability evaluation is not the main content in the paper. A framework of the review of the research topic is constructed in the paper, but not the framework of rural spatial sustainability evaluation.

Highlights

The first and last highlights are hardly highlights.

Primary Review Analysis

The number a and b of figure 3 are mixed.

L186 The result is as follows (Fig.4).The result for what? The study area or the institutions of the authors?

I think there’s only some of the publications of authors showning in Table 2.

Detail Review Analysis

L249 The evaluation criteria of social, economic, and environmental content is not 3 levels but 3 categories.

Fig10 The starting date of the chart should set to 2010. The chart before 2010 should not draw on the figure.

Discussion and Framework

L480 It should be “concentrated in developing countries or developing regions in developed countries.”

Author Response

Many thanks to the reviewers’ valuable comments and suggestions to improve this manuscript. Please see the attachment for responses to your comments.

Reviewer 2 Report

I have some comments on a paper titled “Review and framework construction of rural spatial sustainability evaluation using multi-criteria decision-making analysis”. Although the paper topic is quite interesting, and this paper has the potential to contribute to the literature, some revisions are required to strengthen the paper’s content.

Specific comments:

  1. For methodology, I guess that author(s) use the PRISMA framework (Liberati et al., 2009; Page et al., 2021) to identify the targeted paper for your analysis. If so, please cite the reference appropriately. Otherwise, please explain why you used the method presented in your paper and cite the reference appropriately.
  2. Many parts of the paper are required to improve to make your writing more clear. For example, in Figure 4. “times of citations” should be directly put in the table rather than using the abbreviation “TC”. The same comment for “Number of papers”, etc.
  3. For the Discussion section, please add “the Limitation Section” to not only acknowledge the shortcomings of the paper but also highlight directions for the follow-up studies for other scholars.
  4. Please improve/elaborate on the contribution of your paper to the literature.
  5. Please consider using the following references for your revising:

Liberati, A., Altman, D. G., Tetzlaff, J., Mulrow, C., Gøtzsche, P. C., Ioannidis, J. P. A., Clarke, M., Devereaux, P. J., Kleijnen, J., & Moher, D. (2009). The PRISMA statement for reporting systematic reviews and meta-analyses of studies that evaluate health care interventions: explanation and elaboration. In Journal of clinical epidemiology (Vol. 62, Issue 10). https://doi.org/10.1016/j.jclinepi.2009.06.006

Page, M. J., McKenzie, J. E., Bossuyt, P. M., Boutron, I., Hoffmann, T. C., Mulrow, C. D., Shamseer, L., Tetzlaff, J. M., Akl, E. A., Brennan, S. E., Chou, R., Glanville, J., Grimshaw, J. M., Hróbjartsson, A., Lalu, M. M., Li, T., Loder, E. W., Mayo-Wilson, E., McDonald, S., … Moher, D. (2021). The PRISMA 2020 statement: an updated guideline for reporting systematic reviews. Systematic Reviews, 10(1), 1–11. https://doi.org/10.1186/s13643-021-01626-4

Author Response

(The authors gave the same response as above.)

Reviewer 3 Report

Thanks for sharing the manuscript titled “Review and framework construction of rural spatial sustainability evaluation using multi-criteria decision-making analysis.” 

Below are my suggestions:

  • You should use words to express numbers below 10 and use numerals when representing numbers 10 and above.
  • You must correct grammatical errors and misspellings. For instance, foci are plural of focus, not focuses on line 24. This line must be “On this basis, this study finally proposes five research foci  in the future.” Chapter(s) are wrong word choices. The following must be sections in 1.3. Research purpose and scope. Hereafter is a better word choice than hereinafter on line 75. “Summarize the framework system of existing research.” (line 127) is not a sentence. These are a few of many. Please correct all of them. 
  • Figures and their descriptions can help readers understand the manuscript better and easier. However, too many figures reverse the effectiveness. There are 19 figures, and I could not understand them all. Please keep those only essential. Also, you must describe each figure completely so readers understand it. Are they sourced from you or somebody else? You need to note it. 
  • Why do you need highlights? Your abstract must include your research highlights. Neither does this help understand the manuscript better.

Construction Issues:

  • This manuscript is tough to follow. Yes, a brief introduction is preferred to get to the point. However, the background does not suffice to lead the rest of the sections. How do you define sustainable rural development? This is big and broad. Everyone talks about sustainability nowadays. How is this important to your study? Do you focus on only China, a few countries, or the world? I can find thousands of “rural sustainable development” articles on Google Scholar. You also found 1,040 articles from three databases, but only 103 articles remain for your study. Considering this big, broad principle, it is hard to believe only 103 articles are tied to your research. Yes, “spatial” can narrow your findings, but remember that almost all research articles study regions/areas. I am not convinced enough of your conclusions based on 103 articles on sustainable rural development.     
  • Databases: You must describe all three databases. Also, why did you choose those databases? 
  • Methods: You used multiple methods/criteria, and I am unfamiliar with them. You must thoroughly explain every method or criterion to other prospective readers unfamiliar with them and me. What are subjective and objective weights? How are they different?  
  • Research Questions: The ones on lines 122-127 are barely research questions. They must be specific and connected to your results and conclusions. 
  • Research flow: There is a ton of information in the manuscript and spread out everywhere. They are disconnected. Why is section 5.2. Methods and procedures located at almost the end of the manuscript?   
  • Literature Review: I understand the principal value of this manuscript is to review the relevant literature. And you should have a separate section of the literature review to incorporate the relevance of your research. 
  • Results and Conclusions: There is no Results section. What are those, and how do they help answer your research questions? Are conclusions connected to the precedents? 

Author Response

(The authors gave the same response as above.)

Round 2

Reviewer 2 Report

Thank you for carefully addressing my comments/suggestions. I have no further comments on your paper.

Author Response

Thanks again to the reviewers for their detailed review of the manuscript. We will further check the manuscript information and citations to ensure the correctness of the article.

Reviewer 3 Report

Thank you for the major revision. I believe the current version is well-read and more aligned with the context. An extensive literature review is connected to a volume of information to misleading readers; the revision handles this potential pitfall well. 

Please check with me for a few minor changes. First, you do not need to include the authors’ first name initials in the text. Please keep only last names, e.g., Gebre et al. [27] provided… in line 70. You can also write Rahmoun et al. [141] starting from… in line 348. For more than one author, you can include the last name and et al. Second, you wrote “stock development” in the abstract and “stock era” in line 537. What do you mean by them? Third, please spell out UNESCO in the text. Fourth, please ensure all the citations in the text and reference pages.

Author Response

Many thanks to the reviewers’ valuable comments and suggestions to improve this manuscript. Please see the attachment for the responses to comments.
